# Comparison of Oxidative and Hypoxic Stress Responsive Genes from Meta-Analysis of Public Transcriptomes

**DOI:** 10.3390/biomedicines9121830

**Published:** 2021-12-03

**Authors:** Takayuki Suzuki, Yoko Ono, Hidemasa Bono

**Affiliations:** Program of Biomedical Science, Graduate School of Integrated Sciences for Life, Hiroshima University, 3-10-23 Kagamiyama, Higashi-Hiroshima, Hiroshima 739-0046, Japan; m216692@hiroshima-u.ac.jp (T.S.); d205302@hiroshima-u.ac.jp (Y.O.)

**Keywords:** oxidative stress, RNA-seq, meta-analysis, hypoxia

## Abstract

Analysis of RNA-sequencing (RNA-seq) data is an effective means to analyze the gene expression levels under specific conditions and discover new biological knowledge. More than 74,000 experimental series with RNA-seq have been stored in public databases as of 20 October 2021. Since this huge amount of expression data accumulated from past studies is a promising source of new biological insights, we focused on a meta-analysis of 1783 runs of RNA-seq data under the conditions of two types of stressors: oxidative stress (OS) and hypoxia. The collected RNA-seq data of OS were organized as the OS dataset to retrieve and analyze differentially expressed genes (DEGs). The OS-induced DEGs were compared with the hypoxia-induced DEGs retrieved from a previous study. The results from the meta-analysis of OS transcriptomes revealed two genes, *CRIP1* and *CRIP3*, which were particularly downregulated, suggesting a relationship between OS and zinc homeostasis. The comparison between meta-analysis of OS and hypoxia showed that several genes were differentially expressed under both stress conditions, and it was inferred that the downregulation of cell cycle-related genes is a mutual biological process in both OS and hypoxia.

## 1. Introduction

Oxidative stress (OS) is characterized by an imbalance between oxidants and antioxidants, caused by an increase in the levels of reactive oxygen species (ROS) in a biological system. ROS comprise free radicals that can damage cellular molecules and disrupt homeostasis when antioxidants are downregulated, or ROS levels are upregulated. Chronic OS has been observed in various diseases such as Parkinson’s disease, hepatitis, and cancer [1,2,3,4,5]. 

Due to its strong relationship with human health, the mechanisms of OS have been extensively investigated to provide biological and medical knowledge. These include the mechanism of DNA damage by the highly reactive hydroxyl radicals, the role of OS in the appearance of carcinogenesis and the increase in OS-inducible inflammatory cells by activation of specific transcription factors such as NF-E2-related factor-2 (NRF2) [6,7]. The past studies have also resulted in 435 genes in *Homo sapiens* annotated with the term “GO:0006979 response to oxidative stress” in gene ontology (GO). On the other hand, the broadness of OS-inducible factors and the dynamics of ROS in biological systems make the OS studies challenging and complicated [8]. Despite attempts to list and categorize the OS-related compounds, contributing factors for OS involve an enormous range of both external and internal sources [1] and distinguishing oxidative and non-oxidative sources is challenging. Therefore, the present study focused on analyzing the common features among various sources of OS from the perspective of changes in gene expression. As for another underdeveloped area of OS studies, a clear line between other types of stresses and OS has not been defined. It is necessary to compare OS and other stresses such as hypoxia, which is also an oxygen-related stress condition.

Hypoxia is characterized by reduced oxygen availability in tissues and is known to increase ROS levels through changes in signaling cascades and protein expression [9]. A previous study has successfully attained the collective intelligence of public hypoxic transcriptomes by analyzing 944 runs of RNA-seq data [10]. This approach, a statistical analysis of combined results from multiple studies, called meta-analysis, has attracted attention. This is because the data-driven nature of meta-analysis makes it possible to obtain new findings that are difficult to achieve with traditional hypothesis-driven research methods [11]. The dataset and results obtained in the meta-analysis of hypoxia are valuable sources for both hypothesis- and data-driven research. 

To discover novel areas by utilizing valuable open sources, we collected OS transcriptomes of human-cultured cells from public databases and performed a meta-analysis. This study aimed to investigate the key genes and characteristics for both OS and the comparison between OS and hypoxia by analyzing the differentially expressed genes (DEGs) from the meta-analysis of both OS and hypoxia. The investigation was based on 1783 sets of RNA-seq data (839 from this study and 944 from our previous study of meta-analysis in hypoxia [10]). These investigated genes, the curated dataset for OS, and the method described in this study to compare the results of multiple meta-analyses are expected to be valuable sources for promoting future studies.

## 2. Materials and Methods

### 2.1. Curation of Public Gene Expression Data

As a first step to access and view the integrated expression metadata from public databases, we initially used a graphical web tool, All Of gene Expression (AOE) [12]. AOE provides integrated information about gene expression data integrated from Gene Expression Omnibus (GEO) [13], ArrayExpress [14], Genomic Expression Archive [15], and RNA-seq data only archived in the Sequence Read Archive (SRA) [16]. Extensive keywords, including oxidative stress, rotenone, paraquat, hydrogen peroxide (H_2_O_2_), UV, lipopolysaccharide, arsenite, and deoxynivalenol, were searched in GEO to collect a list of experiment data series related to the RNA-seq data of OS in humans. Then, we manually curated the adequate data with four main criteria: total RNA or polyA-RNA for extracted molecules (sequencing library type), relation to the definition of oxidative stress, relation to an increase in the ROS level, and availability of the corresponding control data (normal state) to pair with the OS data. 

### 2.2. RNA-seq Data Retrieval, Processing, and Quantification

We used Ikra for RNA-seq data retrieval, processing, and quantification. Ikra is an automated pipeline program for RNA-seq data of *Homo sapiens* and *Mus musculus* [17]. Ikra automates the following processes: conversion of the collected SRA format data to FASTQ formatted files using fasterq-dump (version.2.9.6) [18], quality control and trimming of transcript reads with trim-galore (version 0.6.6) [19], and quantification of the transcripts in a unit of transcripts per million (TPM) by salmon (version 1.4.0) [20] with reference transcript sets in GENCODE Release 31 (GRCh38.p12). 

### 2.3. Calculation of ON_ratio and ON_score

We calculated the ratio of expression value of each gene in all pairs between Oxidative stress and Normal state (termed as ON_ratio) [10,11]. Biological replicates from the same data series were treated as individual experiments. The ON_ratio was calculated using Equation (1):(1)$$\mathrm{ON}\_\mathrm{ratio}=\frac{T_{OS}+1}{\;T_{normal\;state}+1}$$

T corresponds to the expression value quantified in TPM. A small number (1 in this case) was added to the expression value to avoid the calculation of zero. ON_ratio values helped to classify each gene into three groups: upregulated, downregulated, and unchanged. When the ON_ratio was greater than the threshold, the gene was considered upregulated, and when the ON_ratio was less than the threshold, the gene was treated as downregulated, otherwise the gene was categorized as unchanged. We adopted 5- and 10-fold thresholds for upregulation and 0.2- and 0.1-fold thresholds for downregulation after testing several thresholds.

To take all the collected RNA-seq data pairs into account, we calculated an Oxidative stress Normal state score (termed as ON_score [11]) based on ON_ratio values using Equation (2): ON_score = count number_upregulated_ – count number_downregulated_(2)

ON_ratio and ON_score were previously introduced in the meta-analysis of OS transcriptome in insects [11] and the meta-analysis of hypoxic transcriptome [10] (termed as HN-ratio and HN-score in the meta-analysis of hypoxia).

### 2.4. Analysis and Comparison of Gene Sets

Differentially expressed gene sets were analyzed by using the web tool Metascape (https://metascape.org/, accessed on 20 September 2021) [21], which performs gene set enrichment analysis. We examined the corresponding terms and p-values obtained using the gene set enrichment analysis. We also used a web Venn diagram tool [22] to search and visualize the matched genes among different gene sets. 

## 3. Results

### 3.1. Data Curation/Collection of Oxidative Stress Transcriptome Data

We collected 839 sets of RNA-seq data and curated them as the OS dataset with 386 pairs of OS and normal state transcriptome data. As OS is caused by various factors, sources of OS in the OS dataset include hydrogen peroxide (H_2_O_2_), UV, rotenone, lipopolysaccharide, arsenite, radiation, *NRF2* knockdown/KO, *BRD4* KO, deoxynivalenol, palmitate, cadmium, methylmercury, zinc dimethyldithiocarbamate, aging, paraquat, and others (Table 1). The proportion of the data pairs of hydrogen peroxide, UV, and rotenone against the total 386 pairs was as follows: 25%, 15%, and 12%, respectively. The percentage of the samples derived from cancer cells was 18% (71 pairs out of 386 pairs). Other metadata about the OS dataset such as each SRA project ID, SRR ID, cell type, concentration of treatment, hours of treatment, and library type of sequencing are shown in Figshare [23].

### 3.2. Verifying the Characteristics of DEGs Using the OS Dataset

A schematic view of the analysis is shown in Figure 1. The most upregulated 493 genes and the most downregulated 492 genes, in a total of 985 genes (5% of the total coding genes in GENCODE Release 31 (GRCh38.p12)), were retrieved by ON_score 10 as DEGs. We performed gene set enrichment analysis using Metascape to visualize the characteristics of the DEGs. The analysis showed that the 493 most upregulated genes by OS were related to “GO:0009617: response to bacterium” and “M5885: NABA matrisome associated” (Figure 2a). The 492 most downregulated genes by OS were related to “GO:0000280 nuclear division” and “R-HAS-69278: Cell Cycle, Mitotic” (Figure 2b). We then found that 32 out of 985 genes were common to genes annotated with GO:0006979 (response to oxidative stress). The most upregulated genes common to GO annotation were *IL6*, *PTGS2*, *and MMP3*, and the most downregulated genes common to GO annotation were *CDK1*, *SELENOP*, and *KLF2* (Figure 2c). The same procedure to verify the DEGs retrieved by ON_score 5 was also performed [23]. The use of ON_score 5 reveals a gene set that includes genes not as differentially expressed as ON_score 10. This shows the broader characteristics of the OS. We used ON_score 5 in the analysis of Section 3.4.

### 3.3. Evaluation of DEGs by Oxidative Stress

To evaluate the genes exceptionally expressed by OS, the parameter of ON_score 10 was applied to retrieve 985 DEGs. Thirty-two genes that were already annotated with GO:0006979 (response to oxidative stress) were excluded from the DEGs, thus revealing OS-related genes which had not yet attracted attention (Figure 1a). The most upregulated 10 genes and the most downregulated 10 genes were retrieved and analyzed (Figure 1a and Figure 3). Five out of the ten most downregulated genes (*H2BC14*, *PIMREG*, *KIF20A*, *CDC20*, and *H2AC14*) were related to the cell cycle. Two of them (*H2BC14* and *H2AC14*) encode the core components of histones. In addition, two genes encoding zinc binding domains (*CRIP1* and *CRIP3*) are included in the list of the ten most downregulated genes. In contrast, the three most upregulated genes were *CCL20*, *CXCL8*, and *CXCL1*, encoding C-C motif chemokine-20, interleukin-8, and growth-regulated alpha protein, respectively. Genes that respond to inflammation were included in the most upregulated genes.

### 3.4. Comparison of the Meta-Analysis Results by OS and Hypoxia

Schematic descriptions of the retrieval and analysis of the downregulated genes in both OS and hypoxia are shown in Figure 1b. We collected 985 DEGs of OS and hypoxia using the ON_score and HN-score. Each set of DEGs was divided into two gene sets: the 493 most upregulated genes and the 492 most downregulated genes. The four gene sets derived from the two types of stress conditions were compared using Venn diagrams to show the common differentially expressed genes (Figure 4a). We found that 44 genes were upregulated in both stress conditions (termed as HN_up ON_up), 50 genes were downregulated in both stress conditions (termed as HN_down ON_down), 11 genes were upregulated in hypoxia but downregulated in OS (termed as HN_up ON_down), and 8 genes were upregulated in OS but downregulated in hypoxia (termed as HN_down ON_up). The number of genes upregulated or downregulated in both stress conditions was greater than the number of genes upregulated or downregulated under either one of the stress conditions. 

The characteristics of each gene set in common were analyzed by performing gene set enrichment analysis using Metascape. “R-HAS-69278: Cell Cycle, Mitotic” and “GO:1903047: mitotic cell cycle process” are the most enriched terms with log_10_(*p*-value) of −19.21 and −18.93 for HN_down ON_down (Figure 4b). HN_up ON_up is related to the terms “M145: PID P53 Downstream pathway” and “M166: PID ATF2 pathway” (Figure 4c). HN_up ON_down and HN_down ON_up included 11 genes and 8 genes, respectively. A list of genes in each gene set is shown in Figshare [23]. 

## 4. Discussion

In this study, we curated the 386 pairs of OS-related RNA-seq data collected from public databases. The collected data were systematically processed and analyzed to identify the DEGs related to OS. Gene set enrichment analysis was performed to identify and confirm the characteristics of the DEGs. In addition, we implemented a new approach to analyze the relationship between the two types of stresses, OS and hypoxia, by comparing the results of both meta-analyses [10]. We compared the genes upregulated and downregulated by hypoxia and OS to obtain four new gene sets, HN_up ON_up, HN_down ON_down, HN_up ON_down, and HN_down ON_up. Each gene set was analyzed using gene set enrichment analysis. 

Meta-analysis of the OS dataset revealed two interesting genes encoding cysteine-rich proteins (*CRIP1* and *CRIP3*) that were the 10th and 5th most downregulated by OS, respectively. Each encoded protein contains zinc-binding domains, and the protein encoded by *CRIP1* is considered to act as a zinc transporter and absorption agent [24,25]. Previous studies have reported several roles for zinc in antioxidant defense systems. For example, zinc inhibits the enzyme nicotinamide adenine dinucleotide phosphate oxidase (NADPH-Oxidase) and promotes the synthesis of metallothionein which contributes to the reduction in ROS [26]. Zinc is also known as a component of the enzyme superoxide dismutase (SOD) which acts to reduce and maintain ROS levels in cells [26]. On the other hand, excess zinc exhibits other toxicities leading to symptoms such as nausea, vomiting, fever, and headaches [27]. Therefore, zinc homeostasis is one of the key biological systems for preventing various types of stress. As the proteins encoded by *CRIP1* and *CRIP3* contain zinc-binding domains, we can assume that they participate in the regulation of zinc homeostasis. Based on this hypothesis and the results of this study, we suggest that the regulation of zinc homeostasis is impaired in OS due to decreased expression of *CRIP1* and *CRIP3*. Since zinc deficiency is known to be a cause of OS [3,28], we speculate that the downregulation of *CRIP1* and *CRIP3* is affected by OS-induced pathways that contribute to the reduced availability of zinc in cells. Uncovering the functions of *CRIP1* and *CRIP3* could be a way to clarify some of the relationships between OS and zinc homeostasis, which may promote the development or the prevention of OS and zinc homeostasis-related diseases such as atherosclerosis [29], Parkinson’s disease [30], cancer, and hepatitis virus infection [31,32].

Comparing the meta-analysis results by two types of stresses, OS and hypoxia, revealed gene sets that were found to be differentially expressed in both stresses. Particularly the gene set downregulated in both stresses showed distinct characteristics with the cell cycle (Figure 4b). This result supports the previous biological findings that DNA damage induced by increased ROS levels causes cell cycle arrest or apoptosis [33,34]. In addition, an increase in ROS production in mitochondria is known to be a common event in both OS and hypoxia [35]; therefore, the downregulation of cell cycle-related genes was an expected result. Furthermore, a meta-analysis of the OS dataset revealed five cell cycle-related genes—*H2BC14*, *PIMREG*, *KIF20A*, *CDC20*, and *H2AC14*—that were, respectively, 2nd, 3rd, 6th, 7th, and 9th most downregulated by OS, supporting the above observation by showing that DEGs associated with OS are related to the cell cycle. As these ten OS-induced downregulated genes were not included in the genes common to hypoxia, further research is needed to clarify whether the expression of these genes is unique to OS or shared by types of stresses other than hypoxia.

The results of this study may play a role in elucidating the causative mechanisms and development of treatments for such diseases as atherosclerosis (OS- and zinc homeostasis-related), chronic kidney disease, and metabolic syndrome (both OS- and hypoxia-related) [36,37] through further studies on the functions of the important genes revealed here. Utilization of real-time reverse transcription polymerase chain reaction (RT-PCR) can be an effective way to confirm the results from the meta-analysis, to give an example of potential further studies [38]. As the quantity of public expression data increases, the more accurate and detailed information about genes that respond to OS can be obtained by updating the OS dataset in the future. We have also shown the possibility of revealing information about the relationships between the types of stresses by comparing the results from the meta-analysis. Thus, the use of collective intelligence, including the results of this study, which will continue to be produced in the future, makes it possible to efficiently promote studies on the search for key pathways, for causes of diseases, and treatments of diseases. 

## Figures and Tables

**Figure 1 biomedicines-09-01830-f001:**
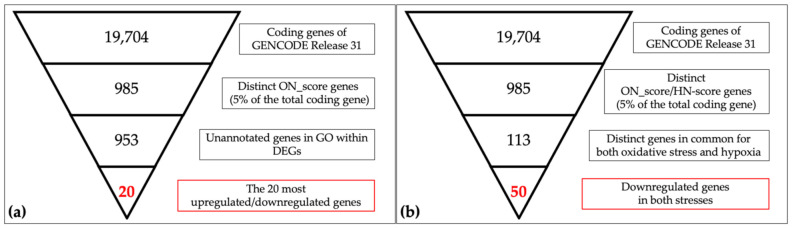
Schematic views of narrowing down the genes in oxidative/hypoxic transcriptome meta-analysis. (**a**) The 19,704 coding genes indexed for the reference genome were filtered by ON_score and by excluding Gene Ontology (GO) annotated genes to retrieve the 20 most differentially expressed genes (DEGs). (**b**) The number of genes downregulated in oxidative stress and hypoxia was then obtained as per the schematic in the figure.

**Figure 2 biomedicines-09-01830-f002:**
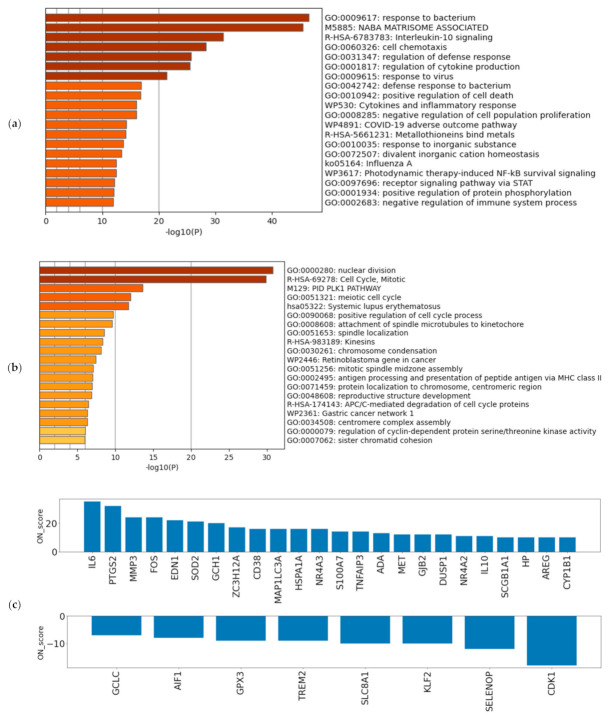
Verifying the characteristics of differentially expressed genes (DEGs): Enrichment analysis for (**a**) the 493 most upregulated genes by oxidative stress (OS) and (**b**) the 492 most downregulated genes by OS. The darker the bar is colored, the more significant the *p*-value. (**c**) ON_score for 32 genes that were identified as DEGs and annotated as GO:0006979 (response to oxidative stress).

**Figure 3 biomedicines-09-01830-f003:**
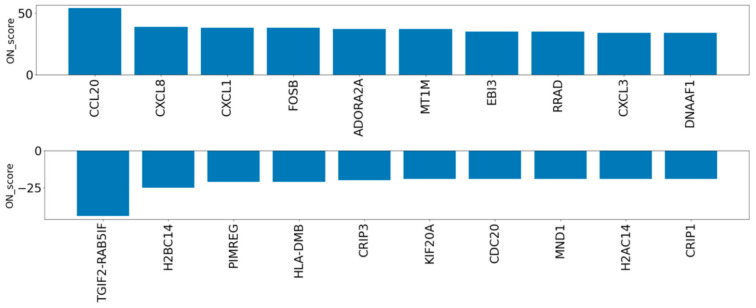
ON_score for the ten most upregulated and downregulated genes after extraction of annotated genes with GO:0006979 (response to oxidative stress).

**Figure 4 biomedicines-09-01830-f004:**
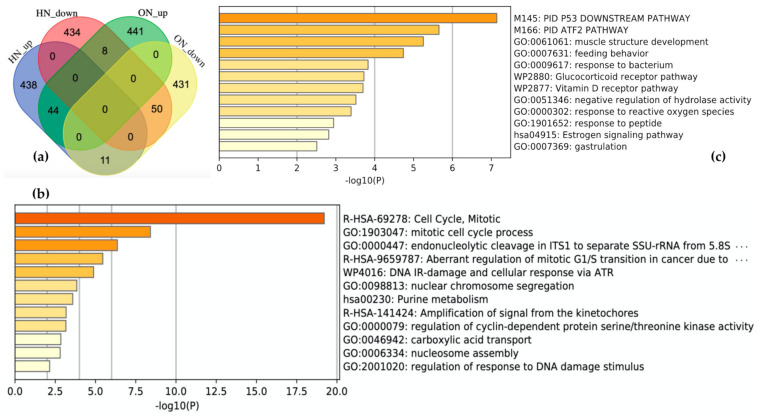
Comparison of results from the meta-analysis in oxidative stress (OS) and hypoxia. (**a**) Visualization of comparison among gene sets. HN_up: the 493 most upregulated genes by hypoxia; HN_down: the 492 most downregulated genes by hypoxia; ON_up: the 493 most upregulated genes by OS; ON_down: the 492 most downregulated genes by OS. Enrichment analysis for (**b**) showed 50 genes downregulated in both stresses and (**c**) 44 genes upregulated in both stresses. The darker the bar is colored, the more significant the *p*-value.

**Table 1 biomedicines-09-01830-t001:** The number of data pairs retrieved for each source of OS.

Source of OS	Number of Data Pairs
Hydrogen peroxide (H_2_O_2_)	98 (25%)
Ultra-Violet rays (UV)	59 (15%)
Rotenone	45 (12%)
Lipopolysaccharide (LPS)	38 (10%)
Arsenite	33 (9%)
Infra-Red rays (Radiation)	24 (6%)
*NRF2* knockdown/KO, *BRD4* KO	22 (6%)
Deoxynivalenol	10 (3%)
Palmitate/high fat/high glucose	10 (3%)
Cadmium, Methylmercury, Zinc dimethyldithiocarbamate	8 (2%)
Aging	6 (2%)
Paraquat	5 (1%)
Others (Senescence, Menadione, entinostat, etc.)	28 (7%)
Total	386

## Data Availability

The data presented in this study are publicly available in Figshare [23].

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
