# Peer review of "Comparison of Oxidative and Hypoxic Stress Responsive Genes from Meta-Analysis of Public Transcriptomes"

_biomedicines, 2021, doi:10.3390/biomedicines9121830_

Round 1

Reviewer 1 Report

In this study, authors perform meta-analysis using public RNA-seq datasets to compare the stress response genes under oxidative and hypoxic stresses. Overall, the story is well organized and the writing is clear. This study will improve our better understanding of gene regulatory network under oxidative and hypoxic stresses. It will benefit the manuscript if some concerns can be addressed.

  1. For the paired RNA-seq datasets (normal vs stress), are there biological replicates for each paired sample? If yes, the normalization between samples, not just the TPM calculated by SALMON, should be performed.
  2. The sequencing library types of RNA-seq datasets used in this study should be provided: total RNA-seq, or polyA+ RNA-seq, ribosome-depletion RNA-seq, or others? For different library types, the data processing is different. For example, the reads from ribosomal RNAs and other constitutive non-coding RNAs (e.g., tRNA) should be removed if using total RNA-seq in case that these extremely-abundant non-coding RNAs affect the differential gene expression analysis when only TPM was applied for normalization. Please refer the paper “Conesa, A., et al. (2016). "A survey of best practices for RNA-seq data analysis." Genome Biol 17(1): 13.”
  3. In Figure 2a, 2b, 4a and 4b, what does the color range of the column mean?
  4. In Figure 2c and 3, the Y-axis label should be given even though it can be learnt from the legend.

Reviewer 2 Report

Dear all, i regret to inform you that this paper is unsuitable for publication here. Although the authors performed adequately the metanalysis i don't think they worth publication in such a high IF journal. They could have used real time experiments in randomly selected individuals to confirm their results by RT PCR analysis but they didn't.Additionally the MS is too short. I suggest them to resubmit as a short communication research into another journal
